# Visible-light photoredox-catalyzed C–O bond cleavage of diaryl ethers by acridinium photocatalysts at room temperature

Fang-Fang Tan[1], Xiao-Ya He[1], Wan-Fa Tian[1] & Yang Li [1,2✉]

Cleavage of C–O bonds in lignin can afford the renewable aryl sources for fine chemicals. However, the high bond energies of these C–O bonds, especially the 4-O-5-type diaryl ether C–O bonds (~314 kJ/mol) make the cleavage very challenging. Here, we report visible-light photoredox-catalyzed C–O bond cleavage of diaryl ethers by an acidolysis with an aryl carboxylic acid and a following one-pot hydrolysis. Two molecules of phenols are obtained from one molecule of diaryl ether at room temperature. The aryl carboxylic acid used for the acidolysis can be recovered. The key to success of the acidolysis is merging visible-light photoredox catalysis using an acridinium photocatalyst and Lewis acid catalysis using Cu (TMHD)$_2$. Preliminary mechanistic studies indicate that the catalytic cycle occurs via a rare selective electrophilic attack of the generated aryl carboxylic radical on the electron-rich aryl ring of the diphenyl ether. This transformation is applied to a gram-scale reaction and the model of 4-O-5 lignin linkages.

[1] Frontier Institute of Science and Technology and State Key Laboratory of Multiphase Flow in Power Engineering, Xi'an Jiaotong University, Xi'an, Shaanxi 710054, People's Republic of China. [2] State Key Laboratory of Elemento-Organic Chemistry, Nankai University, Tianjin 300071, People's Republic of China. ✉email: liyang79@mail.xjtu.edu.cn

Lignin is one of the major components of available biomass in nature[1–3]. In lignin, there are three major types of aryl ether bonds of α-O-4 (~218 kJ/mol), β-O-4 (~289 kJ/mol), and 4-O-5 (~314 kJ/mol)[1–3]. Cleavage of C–O bonds in lignin can afford renewable aryl sources for fine chemicals. However, the high bond energies of these C–O bonds, especially the 4-O-5-type diary ether C–O bonds, make the cleavage very challenging[1–10]. Therefore, in fundamental research, the cleavage of aryl C–O bonds has attracted much attention[11–18].

Cleavage of the α-O-4 and β-O-4 types C–O bonds has been studied, even under mild conditions by visible-light photoredox catalysis[19–23]. For cleavage of the 4-O-5-type diary ether C–O bonds, classical studies focused on the hydrolysis by supercritical water, and the hydrogenolysis using the model of 4-O-5 lignin linkages with poor selectivity[2,4,5,10]. Many aryl skeletons were destroyed. In selective cleavage methods, the use of stoichiometric alkali metals[24–26] or electrocatalytic hydrogenolysis[27–29] limited the large-scale applications because of the associated high costs.

In recent years, selective hydrogenolysis under comparatively milder conditions was developed by Hartwig, Grubbs, Wang et al. (Fig. 1a)[30–41]. The hydrogenolysis was accomplished by using [Ni], [Fe], or [Co] as catalyst, with H₂ (1–6 bar) or LiAlH₄ (2.5 equiv.) as reductant at 120–180 °C[30–36]. A higher than the stoichiometric strong base, such as NaO$^t$Bu/KO$^t$Bu/KHMDS (2.5 equiv.), is important for the selectivity (Fig. 1a, conditions A). Without using a transition-metal catalyst, a combination of Et₃SiH/NaH (≥2.5 equiv.) and KHMDS (2.5 equiv.) at 140–165 °C achieved the selective hydrogenolysis (Fig. 1a, conditions B)[37,38]. More importantly, by MOF or Pd/C as catalyst without a base, the selective hydrogenolysis with H₂ (10–30 bar) at 120–200 °C resulted in no more than 55% yields (Fig. 1a, conditions C)[39–41]. In these studies, a large amount of reductant, strong base, and/or the low yield are/is still the limiting factor(s) for the applications.

Thus, there exists a strong incentive to develop more practical methods for diaryl ether C–O bonds cleavage, toward the utilization of lignin as renewable aryl sources.

We developed the C–O bond cleavage of diaryl ethers containing a carboxylic acid group on the *ortho* position by a visible-light photoredox-catalyzed intramolecular aryl migration from an aryl ether to the *ortho* carboxylic acid group at room temperature (rt), and a following one-pot hydrolysis[42]. Thus, we envisioned the possibility of C–O bond cleavage of general diaryl ethers by an aryl acidolysis with an aryl carboxylic acid followed by hydrolysis, which would expand the scope of the special diaryl ethers largely, even to the model of 4-O-5 lignin linkages.

Specifically, in the aryl acidolysis, photoredox catalysis affords an aryl carboxylic radical **A**, then followed by its electrophilic attack on the diphenyl ether, and a single electron transfer (SET) with a proton and an electron (Fig. 2). However, the two issues make the transformation more challenging. First, although some intramolecular electrophilic attack reactions of aryl carboxylic acid radicals to arenes have been reported[42–47], the intermolecular electrophilic attack reaction of aryl carboxylic acid radicals to arenes has not been successfully explored. The intermolecular electrophilic attack of aryl carboxylic acid radicals to

**a**

conditions:

A. cat. [Ni]/[Fe]/[Co], **H₂ (1–6 bar)/LiAlH₄ (2.5 equiv); NaO$^t$Bu/KO$^t$Bu/KHMDS (2.5 equiv); 120–180 °C, ≥16 h.**

B. **Et₃SiH/NaH (≥2.5 equiv); KHMDS (2.5 equiv); 140–165 °C, ≥20 h.**

C. cat. MOF/Pd/C, **H₂ (10–30 bar); 120–200°C, ≥15 h; ≤55%.**

**b**

**electron-withdrawing groups on C9-arene**            X-ray structure of **PC**$^c$

$^c$Thermal ellipsoids are drawn at 30% probability with H atoms and BF₄⁻ omitted for clarity.

● **H₂O instead of reductants**   ● **rt**   ● **two molecules of phenols**   ● **application to the model of 4-O-5 lignin linkages**

**Fig. 1 Selective C–O bond cleavage of diaryl ethers. a** Typically selective hydrogenolysis of diaryl ethers[30–41]. **b** This work: visible-light photoredox-catalyzed acidolysis of diaryl ethers and the following hydrolysis.

**Fig. 2 Designed pathway of visible-light photoredox-catalyzed acidolysis of diaryl ethers with an aryl carboxylic acid.** Photoredox catalysis affords an aryl carboxylic radical (**A**), then followed by its electrophilic attack on the diphenyl ether to afford **B**, and a single electron transfer (SET) with a proton and an electron.

arenes was proposed in the thermal decomposition of substituted dibenzoyl peroxides in diphenyl ether, in which the corresponding aryl benzoates were obtained in less than 39% yields with low selectivity. In addition, in comparison with the substituted dibenzoyl peroxides, the amount of arenes was huge because diphenyl ether was used as solvent[48]. Second, similarly to the intramolecular reaction, the formation of the more stable ester C–O bond should be the driving force of the diaryl ether C–O cleavage, as the C–O bond energies of an aryl ether and an ester are about 78.8 and 87–93 kcal/mol, respectively[49,50]. However, the possible intermediate **B** may lack a stronger driving force of C–O bond cleavage, compared with the six-membered ring intermediates in the intramolecular reactions[42–47]. As mentioned below, less than 10% yields of the product were obtained with the remaining starting material under the optimized reaction conditions for intramolecular reactions (Table 1, entries 1, 2). Perhaps a Lewis acid could activate the aryl ether by coordination of the oxygen atom.

Herein, we report visible-light photoredox-catalyzed C–O bond cleavage of diaryl ethers by an acidolysis with an aryl carboxylic acid and a following one-pot hydrolysis at rt (Fig. 1b). Phenols with some advantages than arenes in transformations such as aminations, the functionalization, cross-coupling reactions[51,52], are obtained. The aryl carboxylic acid used for the acidolysis can be recovered. The key to the success of the acidolysis is merging visible-light photoredox catalysis using an acridinium photocatalyst (PC) and Lewis acid catalysis using Cu(TMHD)$_2$ (Fig. 1b). Inspired by the significant contributions of Fukuzumi[53–55], Nicewicz[56–58], Sparr[59,60] et al.[61–63]. on acridinium PCs, we investigated the use of an acridinium catalyst, in which an aryl ring with electron-withdrawing groups, instead of mesitylene, typically used in other acridinium catalysts[53–63], on the 9-position to give higher efficiency. Furthermore, with this method, the model of 4-O-5 lignin linkages afforded phenol and 2-methoxyphenol in high efficiency. Compared with the developed selective hydrogenolysis in recent years, using H$_2$O instead of a large amount of reductant afforded two molecules of more valuable phenols at rt.

## Results

**Optimization study**. With these considerations in mind, diphenyl ether (**1a**) and 4-methylbenzoic acid (**2a**) were studied as model substrates. Under our developed conditions for the intramolecular C–O bond cleavage[42], using PDI[64] or Acr$^+$-Mes ClO$_4^-$ (**PC 1**)[53–55] as PC, with 10 mol% K$_2$HPO$_4$ as a base, under 450–455 nm blue LEDs irradiation, only <10% yields of phenyl 4-methylbenzoate (**3a**) and phenol (**4a**) were obtained (Table 1, entries 1, 2). Thereafter, a series of Lewis acids such as Cu(OAc)$_2$, Cu(acac)$_2$, Cu(OTf)$_2$, Ni(acac)$_2$, Fe(acac)$_3$, Zn(acac)$_2$, and Cu(TMHD)$_2$ were studied (Supplementary Table 1, entries 1–6 and Table 1, entry 3). Cu(TMHD)$_2$ slightly promoted the

transformation. Adjustment of the wavelength of the blue LEDs to the maximum absorption of **PC 1** (425–430 nm) induced a slightly increased reactivity (Table 1, entry 4). Other solvents such as MeOH, DCE, EtOAc, and acetone did not give any better results (Supplementary Table 1, entries 7–10). As Acr$^+$-Mes ClO$_4^-$ is susceptible to degradation in the presence of oxygen-centered radicals[57], and de-N-methylation is also possible[56], it is deduced that the generated carboxylic acid radical may induce the degradation of Acr$^+$-Mes ClO$_4^-$.

Subsequently, **PC 2** and **PC 3** were tried[57], in which the N-phenyl were used instead of the N-methyl, and also with the *tert*-butyl on the 3- and 6-positions of the acridinium in the latter case. Both factors induced distinct higher efficiencies (Table 1, entries 5 and 6, **3a** in 29 and 55% yields, with **4a** in 22 and 49% yields).

Furthermore, the influence of substitutes on the 9-aryl ring was investigated. Since the complex procedure for the synthesis of **PC 3**[57], **PC 4**–**PC 9**, with a similar skeleton to that of **PC 3** but with different substituents on the 9-aryl ring, by a two-step synthetic procedure[61], were investigated (Table 1, entries 7–12). Notably, the aryl rings with electron-withdrawing groups instead of mesitylene, typically used in other acridinium catalysts[53–63], resulted in noticeably high efficiencies. To our delight, **PC 9** with 2′-Cl and 4′-F on the 9-aryl ring resulted in 80% of **3a** with 71% of **4a** (entry 12). Although the smaller group of 2′-Cl compared with the methyl groups in mesitylene was used, X-ray crystallography of **PC 9** unambiguously confirmed the angle of torsion between the 9-aryl ring and the acridinium ring (Fig. 1b), which is closely related with a longer fluorescence lifetime[53,56]. In addition, a variation of the *tert*-butyl groups from the 2,7-positions to the 3,6-positions resulted in decreased efficiency (entry 13). A further variation of the substituents on the 10-aryl ring revealed that the unsubstituted phenyl gave a slightly higher yield (entry 15). Decreasing the amount of Cu(TMHD)$_2$ to 5 mol % resulted in obviously lower efficiency (entry 16, **3a** in 47% yield with **4a** in 45% yield). Without Cu(TMHD)$_2$, the transformation only gave 12% of **3a** with 7% of **4a** (entry 17). These results indicate the transformation is promoted by Cu(TMHD)$_2$. The amount of base did not influence the reaction efficiency, even without base (entries 18, 19). Other Lewis acids, such as Cu (OAc)$_2$, Cu(acac)$_2$, and Fe(acac)$_2$ instead of Cu(TMHD)$_2$, were investigated once again (entries 20–23), as slight differences during the initial investigation. The distinct promotion of Cu (TMHD)$_2$ (entry 18) was further confirmed in comparison with the results of other Lewis acids (entries 20–23). Without base and Cu(TMHD)$_2$, no reactivity was observed (entry 24). Control experiments indicate that a PC and visible-light irradiation are essential (entries 25, 26). The fluorescence lifetime and the redox potentials of **PC 1**–**PC 12** were determined (Supplementary Table 2). The data do not provide clear insight regarding the higher efficiency achieved using **PC 9**.

**Table 1 Optimization of the reaction conditions.**

| Entry | Lewis acid | PC | Wavelength (nm) | 3a yield(%) | 4a yield(%)[e] |
|---|---|---|---|---|---|
| 1 | — | PDI | 450–455 | <5 | <5 |
| 2 | — | PC 1 | 450–455 | 8 | <5 |
| 3 | Cu(TMHD)$_2$ | PC 1 | 450–455 | 11 | 10 |
| 4 | Cu(TMHD)$_2$ | PC 1 | 425–430 | 15 | 10 |
| 5 | Cu(TMHD)$_2$ | PC 2 | 425–430 | 29 | 22 |
| 6 | Cu(TMHD)$_2$ | PC 3 | 425–430 | 55 | 49 |
| 7 | Cu(TMHD)$_2$ | PC 4 | 425–430 | 46 | 41 |
| 8 | Cu(TMHD)$_2$ | PC 5 | 425–430 | 60 | 50 |
| 9 | Cu(TMHD)$_2$ | PC 6 | 425–430 | 67 | 62 |
| 10 | Cu(TMHD)$_2$ | PC 7 | 425–430 | 74 | 65 |
| 11 | Cu(TMHD)$_2$ | PC 8 | 425–430 | 77 | 67 |
| 12 | Cu(TMHD)$_2$ | PC 9 | 425–430 | 80 | 71 |
| 13 | Cu(TMHD)$_2$ | PC 10 | 425–430 | 63 | 59 |
| 14 | Cu(TMHD)$_2$ | PC 11 | 425–430 | 50 | 48 |
| 15 | Cu(TMHD)$_2$ | PC 12 | 425–430 | 71 | 69 |
| 16[a] | Cu(TMHD)$_2$ | PC 9 | 425–430 | 47 | 45 |
| 17 | — | PC 9 | 425–430 | 12 | 7 |
| 18[b] | Cu(TMHD)$_2$ | PC 9 | 425–430 | 78 | 70 |
| 19[c] | Cu(TMHD)$_2$ | PC 9 | 425–430 | 80 (76)[f] | 70 (69)[f] |
| 20[c] | Cu(OAc)$_2$ | PC 9 | 425–430 | 48 | 38 |
| 21[c] | Cu(acac)$_2$ | PC 9 | 425–430 | 56 | 47 |
| 22[c] | Ni(acac)$_2$ | PC 9 | 425–430 | 13 | 9 |
| 23[c] | Fe(acac)$_2$ | PC 9 | 425–430 | 21 | 19 |
| 24[c] | — | PC 9 | 425–430 | — | — |

Reaction scheme:

1a + 2a (1.2 equiv) $\xrightarrow[\text{blue LEDs (10 W), rt}]{\text{PC (3 mol%), Lewis acid (10 mol%), K}_2\text{HPO}_4 \text{ (10 mol%)}}$ 3a + 4a

PC 1: R$^4$=H, R$^5$=Me, Y=ClO$_4^-$
PC 2: R$^4$=H, R$^5$=Ph, Y=BF$_4^-$
PC 3: R$^4$=$^t$Bu, R$^5$=Ph, Y=BF$_4^-$

PC 4: R$^6$=2′,4′,6′-tri-CH$_3$  PC 7: R$^6$=4′-F
PC 5: R$^6$=3′,5′-di-CH$_3$  PC 8: R$^6$=2′,4′-di-Cl
PC 6: R$^6$=4′-CF$_3$  PC 9: R$^6$=2′-Cl-4′-F

PC 10: R$^7$=4′-$^t$Bu,
PC 11: R$^7$=2′,4′,6′-tri-CH$_3$
PC 12: R$^7$=H

**Table 1 (continued)**

Reaction scheme: **1a** + **2a** (1.2 equiv) → PC (3 mol%), Lewis acid (10 mol%), $K_2HPO_4$ (10 mol%), blue LEDs (10 W), rt → **3a** + **4a**

Photocatalysts:

PC 1: $R^4$=H, $R^5$=Me, $Y$=$ClO_4^-$
PC 2: $R^4$=H, $R^5$=Ph, $Y$=$BF_4^-$
PC 3: $R^4$=$^tBu$, $R^5$=Ph, $Y$=$BF_4^-$

PC 4: $R^6$=2′,4′,6′-tri-$CH_3$   PC 7: $R^6$=4′-F
PC 5: $R^6$=3′,5′-di-$CH_3$       PC 8: $R^6$=2′,4′-di-Cl
PC 6: $R^6$=4′-$CF_3$             PC 9: $R^6$=2′-Cl-4′-F

PC 10: $R^7$=4′-$^tBu$,
PC 11: $R^7$=2′,4′,6′-tri-$CH_3$
PC 12: $R^7$=H

| Entry | Lewis acid | PC | Wavelength (nm) | 3a yield(%) | 4a yield(%)[e] |
|---|---|---|---|---|---|
| 25[c] | Cu(TMHD)$_2$ | | 425–430 | — | — |
| 26[c,d] | Cu(TMHD)$_2$ | PC 9 | 425–430 | — | — |

Reaction conditions: **1a** (0.24 mmol), **2a** (0.20 mmol), **PC** (3.0 mol%), Lewis acid (10 mol%), $K_2HPO_4$ (10 mol%), solvent (2.0 mL), irradiation with blue LEDs (10 W) for 30 h, [1]H NMR yields of **3a** and **4a** were reported by using $Cl_2CHCHCl_2$ as an internal standard.
[a]Cu(TMHD)$_2$ (5 mol%).
[b]$K_2HPO_4$ (5 mol%).
[c]Without base.
[d]In dark.
[e]Due to the volatility of phenol during work-up, phenol was obtained in slightly lower yields than **3a**.
[f]Isolated yield.

**Fig. 3 Substrate scope of carboxylic acids.** Reaction conditions: **1a** (0.50 mmol), **2** (0.60 mmol), **PC 9** (3 mol%), Cu(TMHD)$_2$ (10 mol%), CH$_3$CN (5 mL), irradiation with blue LEDs (425–430 nm, 10 W) for 30 h. Isolated yields were reported. [a]60 h.

**Evaluation of substrate scope.** With the optimized reaction conditions, the substrate scope was investigated (Fig. 3). First, the influence of various substituents on the benzoic acid was investigated. 4-Methoxyl, 4-tert-butyl gave decreased reaction efficiencies (**3ab**, **3ac** in 61–65% yields with **4a** in 56–60% yields). The benzoic acid afforded **3ad** in 55% yield with **4a** in 50% yield. 4-Fluoro, 4-chloro, and 4-bromo induced excellent yields (**3ae**– **3ag** in 87–91% yields with **4a** in 82–84% yields). 4-Nitro, 4-aldehyde also resulted in high efficiencies (**3ah**, **3ai** in 71–77% yields with **4a** in 65–70% yields). The substituents on the 3-position were also studied. Similar to the substituents on the 4-position, fluoro and chloro resulted in high yields (**3ak**, **3al** in 81–86% yields with **4a** in 74–76% yields). Methyl gave a decreased yield (**3aj** in 61% yield). When the methyl was installed on the 2-position, no product was observed. 2-Fluoro resulted in lower efficiency (**3ap** in 63% yield). The phenomenon should be influenced by the steric factor. A series of aromatic heterocycles carboxylic acids containing pyridinyl, furyl, and thienyl were tested. Only thienyl carboxylic acids resulted in moderate efficiencies (**3aq**, **3ar** in 49–54% yields with **4a** in 40–45% yields).

Next, the substituents on the aryl ring of the diphenyl ethers were investigated (Fig. 4). With electron-withdrawing groups such as methyl ester, trifluoromethyl, nitro, cyan, or acetyl on the 4-position of the aryl ring, **3a** (65–74%) and the corresponding phenols with these electron-withdrawing groups (**4b**-**4f**, 62–72%) were selectively obtained in high yields. The results agree with the designed pathway of the electrophilic attack of the generated carboxylic acid radical, including the selective attack on the electron-rich aryl ring of the diphenyl ethers. 4-Bromo afforded two esters **3a** (22%) and **3as** (64%), and two phenols **4g** (21%) and **4a** (54%). The reason for the result may be the synergic effect of induction and conjugation of the bromo. For the 4-phenyl group, **3at** and **4a** were selectively obtained in 84% yields. The

reason for the selectivity may be the 4-phenyl group stabilizing the generated radical intermediate after the electrophilic attack. Under standard conditions, 4-methyl and 4-methoxyl resulted in very low efficiency (<10%). When the reactions were conducted by a stop-flow reactor, 4-methyl resulted in **3au** in 64% yield with **4a** in 54% yield. And 4-methoxyl still resulted in low efficiency with **3av** in 24% yield, which should be caused by the lower oxidation potential of 4-methoxyl-diphenyl ether (+1.39 V vs SCE) than that of the anion of **2a** (+1.45 V vs SCE) to inhibit the aryl carboxylic radical generation (Supplementary Table 3). Methyl, methoxyl, dimethyl substituents on other positions afforded **3aw**–**3ay** in 68–80% yields with comparable yields of **4a**. Symmetric dimethyl, dimethoxyl, dibromo, and dichloro, and asymmetric dichloro on the 3- or 4-positions resulted in good to high efficiencies (**3az**, **3ay**, **3as**, **3aA**, 52–90%, **4h**–**4k**, 50–86%). When the methyl and methoxyl on 2- or 3-position, with 4′-ester, 4′-cynao or 4′-trifluoro, were investigated, the esters with methyl or methoxyl, as well as phenols with these electron-withdrawing groups, were obtained selectively in high yields (**3aw**, **3az**, **3aB**, 72–82%, **4b**, **4e**, **4c**, 70–82%).

**Synthetic application.** To demonstrate the potential application, a gram-scale reaction of **1a** with **2e** in a flow reactor and following one-pot hydrolysis was conducted. **4a** was obtained in 80% yield, with **2e** in 88% recovery rate (Fig. 5a). Meanwhile, the model of 4-O-5 lignin linkage (**1v**)[30] afforded **4a** (71%) and **4l** (75%) in high efficiency, with **2e** in 82% recovery rate (Fig. 5b). A comparatively complex model of 4-O-5 lignin linkage, 2-methoxyl-4-ethyl-2′-methoxyl-5′-methyl diphenyl ether[33], was tested. The transformation was totally inhibited, which should also be caused by its lower oxidation potential (+1.22 V vs SCE) than that of the anion of **2a** (+1.45 V vs SCE) to inhibit the aryl carboxylic radical generation (Supplementary Table 3).

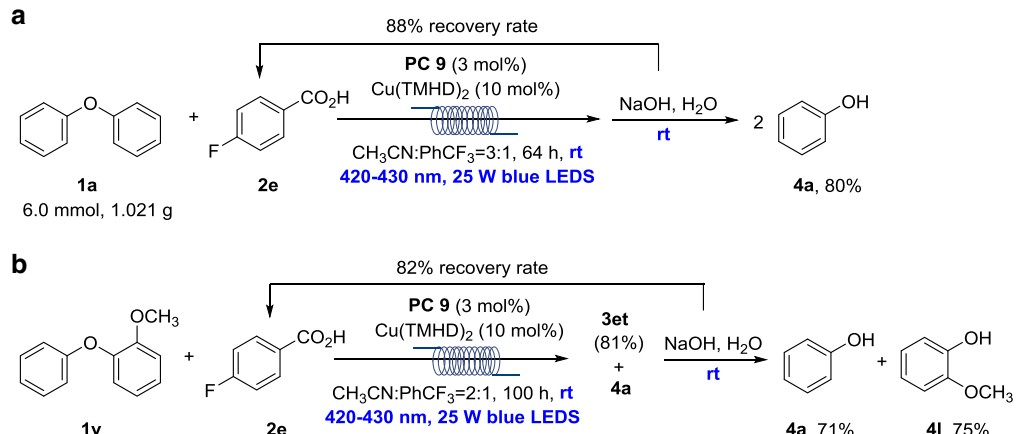

**Fig. 4 Substrate scope of diaryl ethers.** Reaction conditions: **1** (0.5 mmol), **2a** (0.6 mmol), **PC 9** (3 mol%), Cu(TMHD)$_2$ (10 mol%), CH$_3$CN (5 mL), irradiation with blue LEDs (425–430 nm, 10 W) for 30 h. Isolated yields were reported. [a]60 h. [b]CH$_3$CN (5 mL) and PhCF$_3$ (2.5 mL), 60 h. [c]**1** (0.2 mmol), **2a** (0.24 mmol), **PC 9** (3 mol%), Cu(TMHD)$_2$ (10 mol%), CH$_3$CN (2 mL) with PhCF$_3$ (1 mL), stop-flow reactor was used with blue LEDs (420–430 nm, 25 W) irradiation for 60 h.

**Fig. 5 Gram-scale reaction and its application. a** Gram-scale reaction. **b** Application. Due to the difficult purification for the isolated yields of **4a** and **4l**, after the acidolysis, the reaction was worked-up to afford **3et** and **4a** in 81% and 71% yields, respectively.

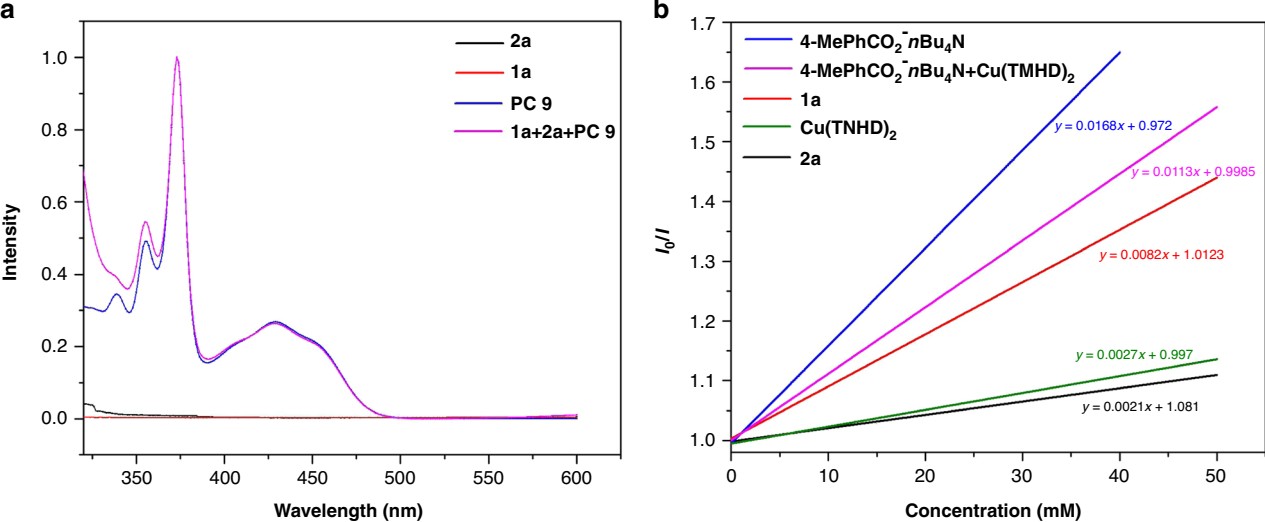

**Fig. 6 UV–vis absorption and fluorescence quenching experiments. a** UV–vis absorption spectra of **1a**, **2a**, **PC 9**, and the reaction mixture. **b** Emission-quenching experiments of the anion of **2a** (4-MePhCO$_2^-$nBu$_4$N$^+$), **2a**, **1a**, Cu(TMHD)$_2$, and the anion of **2a** with Cu(TMHD)$_2$.

**Mechanism studies**. To gain insight into the reaction mechanism, a series of experiments were conducted. First, UV–vis absorption spectra of each component and the reaction mixture confirmed that **PC 9** acts as a PC (Fig. 6a). Second, luminescent quenching experiments were conducted (Fig. 6b). The anion of **2a** (4-MePhCO$_2^-$nBu$_4$N$^+$), **2a**, **1a**, Cu(TMHD)$_2$, and the anion of **2a** with Cu(TMHD)$_2$ quenched the excited-state **PC 9***. The anion of **2a** displayed an obviously larger quenching rate. **1a**, Cu(TMHD)$_2$ displayed lower quenching rates. The anion of **2a** with Cu(TMHD)$_2$ displayed a lower quenching rate in comparison with that of the anion of **2a**. These results suggest there is little possibility for Cu(TMHD)$_2$ to participate in the catalytic cycle of acridinium catalyst. Third, the pH value of the reaction mixture was determined as 3.61 or 4.30, with or without 10 mol% Cu(TMHD)$_2$. Based on these pH values, Cu(TMHD)$_2$ should promote the ionization of **2a** by the salt effect[65]. As mentioned above, the transformation is promoted by Cu(TMHD)$_2$ under base conditions. In addition, under base free conditions, 80% ester **3a** with Cu(TMHD)$_2$ (entry 19), and 13–56% ester **3a** with Cu(OAc)$_2$, Cu(acac)$_2$, Ni(acac)$_2$, Fe(acac)$_2$ (entries 20–23) in comparison with no production of **3a** without any these metal salts and base (entry 23), these results suggest that Cu(TMHD)$_2$ also acts as a Lewis acid to promote the transformation.

Furthermore, with the addition of 2 equiv. of TEMPO as oxidant, compounds **5a** and **5b**, instead of **3** and **4**, were obtained (Fig. 7). This result and the standard reaction suggest that the reaction of **1a** with **2a** generates possible intermediates **B′**, **C**, and **C′** in a reversible manner under the optimized conditions. Under the standard conditions without TEMPO, the diaryl ether bond of **B′** is activated by the Cu$^{2+}$ to promote the equilibrium shifting to give **3a** and **4a** in high efficiency. With TEMPO as oxidant, the equilibrium shifting is promoted by TEMPO to give **5a** and **5b** via **C** and **C′**[57].

The thermodynamic feasibility of the photo-induced SET was analyzed based on the oxidation–reduction potentials. The oxidation potential of $E^{4\text{-}CH3PhCO2\bullet/4\text{-}CH3PhCO2-}$, $E^{1a+\bullet/1a}$, and the reduction potential of $E^{PC\ 9/PC\ 9-\bullet}$ in CH$_3$CN were determined as +1.45 V vs. SCE, +1.86 V vs. SCE[66], and −0.47 V vs. SCE (Supplementary Figs. 11, 12, and 36), respectively. The excited-state energy $E_{0,0}$ of **PC 9** was determined as 2.63 eV (Supplementary Fig. 24). Therefore, the reduction potential of $E^{PC\ 9*/PC\ 9-\bullet}$ was calculated as +2.16 V vs. SCE ($E^{PC*/PC-\bullet} =$

$E^{PC/PC-\bullet} + E_{0,0}$) (Supplementary Fig. 36). These reduction potentials indicate the prior formation of PC$^{-\bullet}$ and the carboxylic acid radical[42–47] by a SET between PC* and the carboxylic acid anion. Based on the electrochemical potentials of phenolic products[66] and **PC 9**, phenolic products can be readily oxidized by **PC 9**. The stability of phenolic products under the reaction conditions may be attribute to the back electron transfer[67,68]. Furthermore, a quantum yield value of φ = 0.20 was determined. Thus, at this stage, whether the reaction proceeds via a photoredox catalytic pathway or a radical chain pathway could not be reached[69].

Based on these results, the reaction mechanism is proposed as shown in Fig. 8. First, Cu(TMHD)$_2$ promotes the ionization of **2a** to afford **2a$^-$** and a proton. Meanwhile, irradiation of PC with blue LEDs leads to the excited-state PC*. A SET occurs between PC* and **2a$^-$** to generate the carboxylic acid radical **A′** and PC$^{-\bullet}$. An electrophilic attack of **A′** occurs on the electron-rich aryl ring of diphenyl ethers to form intermediate **B′**. A SET between **B′** and PC$^{-\bullet}$ in the presence of a proton with the promotion of Cu(TMHD)$_2$ as a Lewis acid affords **3a**, **4a**, with the regeneration of PC.

In summary, we have developed visible-light photoredox-catalyzed C–O bond cleavage of diaryl ethers by an acidolysis and a following one-pot hydrolysis at rt. Two molecules of phenols are obtained from one molecule of diaryl ether in high efficiency. The aryl carboxylic acid used for the acidolysis can be recovered. The transformation is applied to a gram-scale reaction and the model of 4-O-5 lignin linkages. The applications of this approach to more complex models of 4-O-5 lignin linkages and the linkages in native biomass for utilization of lignin as renewable aryl sources are in progress.

## Methods

**General procedure for the C–O bond cleavage of diaryl ethers**. To a quartz tube equipped with a magnetic stirring bar, **PC 9** (0.015 mmol, 3.0 mol%, 9.60 mg), compound **1** (0.50 mmol) compound **2** (0.60 mmol), and Cu(TMHD)$_2$ (0.05 mmol, 10 mol%, 21.5 mg) were added. The tube was evacuated and filled with argon three times with each cycle in 15 min. The freshly distilled solvent was then added into the tube via a syringe under an argon atmosphere, then stirred and irradiated with 425–430 nm blue LEDs at ambient temperature (19–21 °C) in a Wattecs Parallel Reactor (Supplementary Fig. 1) for 30–60 h. After the reaction, the solvent was removed in vacuo and the residue was purified by column chromatography (petroleum ether/EtOAc = 200/1–5/1) to afford compounds **3** and **4**.

Full experimental procedures are provided in the Supplementary Information.

aThe total isolated yield of **5a** and **5b** was 37% with the ratio of 5.6:1.0.

**Fig. 7 Deducing possible intermediate B′a.** With the addition of 2 equiv. of TEMPO as oxidant, compounds **5a** and **5b**, instead of **3** and **4**, were obtained. This result and the standard reaction suggest that the reaction of **1a** with **2a** generates possible intermediate **B′**.

**Fig. 8 Proposed mechanism for the acidolysis.** A selective electrophilic attack of the generated aryl carboxylic radical on the electron-rich aryl ring of the diphenyl ether.

## Data availability

Experimental data, as well as [1]H and [13]C NMR spectra for all new compounds prepared in the course of these studies, are provided in the Supplementary Information file. The X-ray crystallography reported in this study has been deposited at the Cambridge Crystallographic Data Centre (CCDC), under deposition number 2004336. These data can be obtained free of charge from The Cambridge Crystallographic Data Centre via https://www.ccdc.cam.ac.uk/. The data that support the findings of this study are available within the article and its Supplementary Information files. Any further relevant data are available from the corresponding author upon reasonable request. Source data are provided with this paper.

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

## Acknowledgements
This work was supported by the National Key R&D Program of China (Nos: 2018YFB1501601) and the Instrument Analysis Center of the Xi'an Jiaotong University. We thank Prof. Gang He, Xi'an Jiaotong University for the support on cyclic voltammetry experiments and emission-quenching experiments.

## Author contributions
F.-F. T., X.-Y. H., and W.-F. T. performed the experiments and analyzed the data. Y. L. directed the project and wrote the manuscript.

## Competing interests
The authors declare no competing interests.
