## [Peer Review File · Nature Communications]

REVIEWER COMMENTS

Reviewer #1 (Remarks to the Author):

In the paper entitled "Visible-Light Photoredox-Catalyzed C–O Bond Cleavage of Diaryl Ethers by Acridinium Photocatalysts at Room Temperature", Tan, He, Tian and Li describe a powerful visible-light photoredox-catalyzed C–O bond cleavage of diaryl ethers by an acidolysis with an aryl carboxylic acid and a following one-pot hydrolysis. This reaction solves a current challenge in the cleavage of aryl C–O bonds; intramolecular aryl migration from an aryl ether to the ortho carboxylic acid group at rt, and a following one-pot hydrolysis enables synthesis of two phenols with different substituents. The yields are excellent, and the functional group tolerance is also fairly broad. The Supporting Information is thorough and well prepared. I anticipate that the synthetic organic and catalysis communities, will be very excited by this new method. I recommend publication after the following minor revisions have been addressed.

- 1) At least, the outcomes of reactions with heterocycles should be incorporated. Discussion about the heterocycles is necessary.
- 2) Fig 4 is very hard to follow, it should be draw the structure of 4a-4k, so it is easier to read for authors.
- 3) Since excellent yields were found when the electron-withdrawing group on the benzoic acid, so how about these compounds attack on the electron-rich aryl ring of the diphenyl ethers?

Reviewer #2 (Remarks to the Author):

Li et al reported the acridinium-catalyzed C-O bond cleavage of diaryl ethers via intermolecular electrophilic attack of aryl carboxylic acid radicals to arenes. The current conditions for aryl ether cleavage were much milder than previous reports. Besides, two phenols were obtained which are different from the products from typical hydrogenolysis systems. A comprehensive study on acridinium catalysts was displayed. In my opinion, this manuscript could be published after proper revision on discussion of mechanism and substrate scope.

1. Phenolic products can be readily oxidized, and commonly they can act as the radical scavenger. Could the authors discuss the reason of stability of phenolic products in the presence of acridinium catalyst?
2. The copper catalyst was indispensable for the success of ether cleavage. Beyond the possible role as a Lewis acid, could the copper catalyst participate in the catalytic cycle of acridinium catalyst?
3. Could the authors provide some evidence on the interaction between copper catalyst with oxygen atom of diaryl ether? For instance, the effect of Cu(TMHD)₂ to promote the ionization of 2a could be proved by further luminescent quenching experiments in the co-presence of 2a and Cu(TMHD)₂.
4. The model compounds to resemble the native lignin 4-O-5' linkages are too simple. Diaryl ethers simultaneously bearing 2-methoxy, 2'-hydroxy and 4-(4'-) alkyl substituents or their analogues should be tested to prove the potential application in lignin research.
5. Low efficiency was shown in diaryl ethers with 4-methyl and 4-methoxy. However, the methyl and methoxy on 2- or 3-position resulted in good efficiencies. These compounds have similar oxidation potentials, why different efficiencies were shown?
6. TEMPO-captured experiment is quite confusing. The intermediates C could be generated regardless of the addition of TEMPO. Could the intermediate B be detected, or is there some interconversion between intermediate B with C?

Reviewer #3 (Remarks to the Author):

General comments: Lignin is the second most abundant organic matter on Earth, and is an underutilized renewable source for valuable aromatic chemicals. For future sustainable production of aromatic compounds, it is highly desirable to convert lignin into value-added platform chemicals instead of using fossil-based resources. Lignins are aromatic polymers linked by three types of ether bonds (α -O-4, β -O-4, and 4-O-5 linkages) and other C–C bonds. Among the ether bonds, the bond dissociation energy of the 4-O-5 linkage is the highest and the most challenging to cleave. Therefore, it is of great significance to break 4-O-5 diphenyl ether bond and synthesize high value chemicals by cross coupling. To date, 4-O-5 ether linkage model compounds have been cleaved to obtain phenol, cyclohexane, cyclohexanone, and cyclohexanol. In this manuscript, the author have achieved two phenols from diphenyl ether at room temperature through merging visible-light photoredox catalysis with a new acridinium photocatalyst and Lewis acid catalysis with Cu(TMHD)₂. It is a topic of interest to the researchers in the related areas but the manuscript needs minor improvement before acceptance for publication. My detailed comments are as follows:

1. In this manuscript, two phenols were obtained from diphenyl ether under visible photoredox catalysis. Wherein, visible photocatalysis is a method that has been widely studied at present. In this study, the New Acridinium Catalyst, which has been prepared by predecessors, is directly applied. There is no updated contribution in methodological research, but only innovation in the application field.
2. For the above reason, the presentation should be focused on the results. Unfortunately, the material was not properly organized and it is strongly suggested that the authors check carefully the English writing and use standard terminologies in the technical area.
3. In this experiment, 425-430nm blue light is required to be illuminated for 30 hours, which is relatively low in light utilization and the environmental conditions used are not low in cost. Some comparative analysis of energy consumption can be added.
4. The conclusion should be concise and only summarize the most important contribution of the research, and stating what other fields the method can be extended to.

Point-by-point response to the reviewers' comments

REVIEWER COMMENTS

Reviewer #1 (Remarks to the Author):

In the paper entitled "Visible-Light Photoredox-Catalyzed C–O Bond Cleavage of Diaryl Ethers by Acridinium Photocatalysts at Room Temperature", Tan, He, Tian and Li describe a powerful visible-light photoredox-catalyzed C–O bond cleavage of diaryl ethers by an acidolysis with an aryl carboxylic acid and a following one-pot hydrolysis. This reaction solves a current challenge in the cleavage of aryl C–O bonds; intramolecular aryl migration from an aryl ether to the ortho carboxylic acid group at rt, and a following one-pot hydrolysis enables synthesis of two phenols with different substituents. The yields are excellent, and the functional group tolerance is also fairly broad. The Supporting Information is thorough and well prepared. I anticipate that the synthetic organic and catalysis communities, will be very excited by this new method. I recommend publication after the following minor revisions have been addressed.

1) At least, the outcomes of reactions with heterocycles should be incorporated. Discussion about the heterocycles is necessary.

Answer: Thanks a lot for the comment.

In the revised manuscript about the description of the substrate scope, "A series of aromatic heterocycles carboxylic acids containing pyridinyl, furyl and thienyl were tested. Only thienyl carboxylic acids resulted in moderate efficiencies (**3aq**, **3ar** in 49–54% yields with **4a** in 40–45% yields)." was added.

2) Fig 4 is very hard to follow, it should be draw the structure of 4a-4k, so it is easier to read for authors.

Answer: Thanks a lot for the comment.

In the revised manuscript, Fig 4 was revised to make it more easier for authors.

3) Since excellent yields were found when the electron-withdrawing group on the benzoic acid, so how about these compounds attack on the electron-rich aryl ring of the diphenyl ethers?

Answer: Thanks a lot for the comment.

As shown in Figure 5B, benzoic acid **2e** containing fluoro as an electron-withdrawing group reacted with the diphenyl ether **1t** containing methoxy as an electron-donating group to afford **3et** in 81% yield.

Reviewer #2 (Remarks to the Author):

Li et al reported the acridinium-catalyzed C-O bond cleavage of diaryl ethers via intermolecular electrophilic attack of aryl carboxylic acid radicals to arenes. The current conditions for aryl ether cleavage were much milder than previous reports. Besides, two phenols were obtained which are different from the products from typical hydrogenolysis systems. A comprehensive study on acridinium catalysts was displayed. In my opinion, this manuscript could be published after proper revision on discussion of mechanism and substrate scope.

1. Phenolic products can be readily oxidized, and commonly they can act as the radical scavenger. Could the authors discuss the reason of stability of phenolic products in the presence of acridinium catalyst?

Answer: Thanks a lot for the comment.

Based on the electrochemical potentials of phenolic products and the acridinium catalyst, phenolic products can be readily oxidized by the acridinium catalyst. The stability of phenolic products maybe attribute to the back electron transfer, please see the ref. *Angew. Chem. Int. Ed.* **2011**, *50*, 8652–8655. In the description of the mechanism part, “Based on the electrochemical potentials of phenolic products⁶⁶ and **PC 9**, phenolic products can be readily oxidized by **PC 9**. The stability of phenolic products under the reaction conditions maybe attribute to the back electron transfer.^{67,68}” was added.

2. The copper catalyst was indispensable for the success of ether cleavage. Beyond the possible role as a Lewis acid, could the copper catalyst participate in the catalytic cycle of acridinium catalyst?

Answer: Thanks a lot for the comment.

Entry 23 in the original manuscript was adjusted to entry 17 in the revised manuscript to make the description more clear. In the description of optimization of the reaction conditions part, “Without Cu(TMHD)₂, the transformation only gave 12% of **3a** with 7% of **4a** (entry 17). These results indicate the transformation is promoted by Cu(TMHD)₂.” was added in the revised manuscript.

To give some insights whether the copper catalyst participate in the catalytic cycle of acridinium catalyst, we further did the luminescent quenching experiments of the anion of **2a** (4-MePhCO₂⁻ nBu₄N⁺) with Cu(TMHD)₂, and Cu(TMHD)₂. In the revised manuscript, the corresponding description was revised as “The anion of **2a** (4-MePhCO₂⁻ nBu₄N⁺), **2a**, **1a**, Cu(TMHD)₂, and the anion of **2a** with Cu(TMHD)₂ quenched the excited state **PC 9***. The anion of **2a** displayed an obviously larger quenching rate. **1a**, Cu(TMHD)₂ displayed lower quenching rates. The anion of **2a** with Cu(TMHD)₂ displayed lower quenching rate in comparison with that of the anion of **2a**. These results suggest there is little possibility for Cu(TMHD)₂ to participate in the catalytic cycle of acridinium catalyst.”.

3. Could the authors provide some evidence on the interaction between copper catalyst with oxygen atom of diaryl ether? For instance, the effect of Cu(TMHD)₂ to promote the ionization of **2a**

could be proved by further luminescent quenching experiments in the co-presence of **2a** and Cu(TMHD)₂.

Answer: Thanks a lot for the comment.

Indeed, it is very difficult to give the direct evidence of the interaction between copper catalyst with oxygen atom of diaryl ether. We propose Cu(TMHD)₂ as a Lewis acid to activate the diaryl ether on the oxygen atom based on the following experiments.

In the description of optimization of the reaction conditions part, "Decreasing the amount of Cu(TMHD)₂ to 5 mol% resulted in obviously lower efficiency (entry 16, **3a** in 47% yield with **4a** in 45% yield). Without Cu(TMHD)₂, the transformation only gave 12% of **3a** with 7% of **4a** (entry 17). These results indicate the transformation is promoted by Cu(TMHD)₂."

In the description of mechanism studies part, "As mentioned above, the transformation is promoted by Cu(TMHD)₂ under base conditions. In addition, under base free conditions, 80% ester **3a** with Cu(TMHD)₂ (entry 19), and 13–56% ester **3a** with Cu(OAc)₂, Cu(acac)₂, Ni(acac)₂, Fe(acac)₂ (entries 20–23) in comparison with no production of **3a** without any these metal salts and base (entry 23), these results suggest that Cu(TMHD)₂ also acts as a Lewis acid to promote the transformation."

Furthermore, the effect of Cu(TMHD)₂ to promote the ionization of **2a** was proved by the pH values of the reaction mixtures of with (3.61) or without 10 mol% Cu(TMHD)₂ (4.30). In the description of mechanism studies part, it was described as "Third, the pH value of the reaction mixture was determined as 3.61 or 4.30, with or without 10 mol% Cu(TMHD)₂. Based on these pH values, Cu(TMHD)₂ should promote the ionization of **2a** by the salt effect.⁶⁵"

4. The model compounds to resemble the native lignin 4-O-5' linkages are too simple. Diaryl ethers simultaneously bearing 2-methoxy, 2'-hydroxy and 4-(4'-) alkyl substituents or their analogues should be tested to prove the potential application in lignin research.

Answer: Thanks a lot for the comment.

In the synthetic application part, "Comparatively complex model of 4-O-5 lignin linkage, 2-methoxy-4-ethyl-2'-methoxy-5'-methyl diphenyl ether,³³ was tested. The transformation was totally inhibited, which should also be caused by its lower oxidation potential (+ 1.22 V vs SCE) than that of the anion of **2a** (+1.45 V vs SCE) to inhibit the aryl carboxylic radical generation (Table S3)." was added in the revised manuscript.

In the conclusion part, "The applications of this approach to more complex models of 4-O-5 lignin linkages and the linkages in native biomass for utilization of lignin as renewable aryl sources are in progress." was added in the revised manuscript.

5. Low efficiency was shown in diaryl ethers with 4-methyl and 4-methoxy. However, the methyl

and methoxy on 2- or 3-position resulted in good efficiencies. These compounds have similar oxidation potentials, why different efficiencies were shown?

Answer: Thanks a lot for the comment.

We apologize our misleading information in the original manuscript. Under the standard conditions, 4-methyl and 4-methoxy resulted in very low efficiency (<10%). In the revised manuscript, it was revised as "Under standard conditions, 4-methyl and 4-methoxy resulted in very low efficiency (<10%). When the reactions were conducted by a stop-flow reactor, 4-methyl resulted in **3au** in 64% yield with **4a** in 54% yield. And 4-methoxy still resulted in low efficiency with **3av** in 24% yield, which should be caused by the lower oxidation potential of 4-methoxy-diphenyl ether (+1.39 V vs SCE) than that of the anion of **2a** (+1.45 V vs SCE) to inhibit the aryl carboxylic radical generation (Table S3)."

6. TEMPO-captured experiment is quite confusing. The intermediates C could be generated regardless of the addition of TEMPO. Could the intermediate B be detected, or is there some interconversion between intermediate B with C?

Answer: Thanks a lot for the comment.

In the revised manuscript, we revised the Figure 7 to make the points more clear. The description was revised as "Furthermore, with addition of 2 equiv of TEMPO as oxidant, compounds **5a** and **5b**, instead of **3** and **4**, were obtained (Fig. 7). This result and the standard reaction suggest that the reaction of **1a** with **2a** generates possible intermediates **B'**, **C** and **C'** in a reversible manner under the optimized conditions. Under the standard conditions without TEMPO, the diaryl ether bond of **B'** is activated by the Cu^{2+} to promote the equilibrium shifting to give **3a** and **4a** in high efficiency. With TEMPO as oxidant, the equilibrium shifting is prompted by TEMPO to give **5a** and **5b** via **C** and **C'**.⁴⁸".

Reviewer #3 (Remarks to the Author):

General comments: Lignin is the second most abundant organic matter on Earth, and is an underutilized renewable source for valuable aromatic chemicals. For future sustainable production of aromatic compounds, it is highly desirable to convert lignin into value-added platform chemicals instead of using fossil-based resources. Lignins are aromatic polymers linked by three types of ether bonds (α -O-4, β -O-4, and 4-O-5 linkages) and other C-C bonds. Among the ether bonds, the bond dissociation energy of the 4-O-5 linkage is the highest and the most challenging to cleave. Therefore, it is of great significance to break 4-O-5 diphenyl ether bond and synthesize high value chemicals by cross coupling. To date, 4-O-5 ether linkage model compounds have been cleaved to obtain phenol, cyclohexane, cyclohexanone, and cyclohexanol. In this manuscript, the author have achieved two phenols from diphenyl ether at room temperature through merging visible-light photoredox catalysis with a new acridinium photocatalyst and Lewis acid catalysis with $\text{Cu}(\text{TMHD})_2$. It is a topic of interest to the researchers in the related areas but the manuscript needs minor improvement before acceptance for publication. My detailed comments are as follows:

1. In this manuscript, two phenols were obtained from diphenyl ether under visible photoredox catalysis. Wherein, visible photocatalysis is a method that has been widely studied at present. In this study, the New Acridinium Catalyst, which has been prepared by predecessors, is directly applied. There is no updated contribution in methodological research, but only innovation in the application field.

Answer: Thanks a lot for the comment.

Indeed, the general method for the preparation of acridinium catalysts have been prepared and applied. All the “new ” used for the **PC 9** has been deleted.

In our manuscript, we revised as “Inspired by the significant contributions of Fukuzumi,⁴⁴⁻⁴⁶ Nicewicz,⁴⁷⁻⁴⁹ Sparr,⁵⁰⁻⁵¹ et al.⁵²⁻⁵⁴ on acridinium PCs, we investigated the use of an acridinium catalyst, in which an aryl ring with electron-withdrawing groups, instead of mesitylene, typically used in other acridinium catalysts,⁴⁴⁻⁵⁴ on the 9-position to give obviously higher efficiency.”

In the references 44 and 47, the importance of the angle of torsion between the 9-aryl ring and the acridinium ring closely related with a longer fluorescence lifetime was discussed. As the Cl is smaller than the methyl, without X-ray crystallography of **PC 9**, we can not confirm the angle of torsion between the 9-aryl ring. In the X-ray crystallography of **PC 9**, we observed the angle of torsion. Thus during the experiments, we described “Although the smaller group of 2'-Cl compared with the methyl groups in mesitylene was used, X-ray crystallography of **PC 9** unambiguously confirmed the angle of torsion between the 9-aryl ring and the acridinium ring (Fig. 1B), which is closely related with a longer fluorescence lifetime.^{44,47”}.

2. For the above reason, the presentation should be focused on the results. Unfortunately, the material was not properly organized and it is strongly suggested that the authors check carefully the English writing and use standard terminologies in the technical area.

Answer: Thanks a lot for the comment. We carefully polished the English writing and the terminologies. All the revisions were highlighted.

3. In this experiment, 425-430nm blue light is required to be illuminated for 30 hours, which is relatively low in light utilization and the environmental conditions used are not low in cost. Some comparative analysis of energy consumption can be added.

Answer: Thanks a lot for the comment.

The powers of blue LEDs, such as 10 W, 25 W, were added in the Schemes. It can be calculated that 0.3 KW•h was consumed by blue LEDs irradiation for a 30 h reaction by 10 W blue LEDs. We also detected that 0.99 KW•h was consumed for one reaction by using Watecs Blue LEDs Irradiation Parallel Reactor as other parts of the Reactor besides the Blue LEDs also consumed the energy.

4. The conclusion should be concise and only summarize the most important contribution of the

research, and stating what other fields the method can be extended to.

Answer: Thanks a lot for the comment.

The conclusion was revised as "In summary, we have developed visible-light photoredox-catalyzed C–O bond cleavage of diaryl ethers by an acidolysis and a following one-pot hydrolysis at rt. Two molecules of phenols are obtained from one molecule of diaryl ether in high efficiency. The aryl carboxylic acid used for the acidolysis can be recovered. The transformation is applied to a gram-scale reaction and the model of 4-O-5 lignin linkages. The applications of this approach to more complex models of 4-O-5 lignin linkages and the linkages in native biomass for utilization of lignin as renewable aryl sources are in progress."

REVIEWERS' COMMENTS

Reviewer #1 (Remarks to the Author):

I recommend publication after the minor revisions have been addressed.

Reviewer #2 (Remarks to the Author):

The authors have revised the manuscript according to the reviewer's comments, and explained the concerns in details. Therefore, this manuscript is recommended for publication now.

Reviewer #3 (Remarks to the Author):

All the concerns have been well addressed and the manuscript is ready to be accepted.